# Effects of temperature-dependent $NO_x$ emissions on continental ozone production

Paul S. Romer[1], Kaitlin C. Duffey[1], Paul J. Wooldridge[1], Eric Edgerton[2], Karsten Baumann[2,3], Philip A. Feiner[4], David O. Miller[4], William H. Brune[4], Abigail R. Koss[5,6,7], Joost A. de Gouw[6,7], Pawel K. Misztal[8], Allen H. Goldstein[8,9], and Ronald C. Cohen[1,10]

[1]Department of Chemistry, University of California Berkeley, Berkeley, CA, 94720, USA.
[2]Atmospheric Research and Analysis Inc., Cary, NC, 27513, USA.
[3]Department of Environmental Sciences and Engineering, University of North Carolina at Chapel Hill, Chapel Hill, NC, 27599, USA.
[4]Department of Meteorology and Atmospheric Science, The Pennsylvania State University, University Park, PA, 16802, USA.
[5]NOAA Earth System Research Laboratory (ESRL), Chemical Sciences Division, Boulder, CO, 80305, USA.
[6]Cooperative Institute for Research in Environmental Sciences, University of Colorado Boulder, Boulder, CO, 80309, USA.
[7]Department of Chemistry and Biochemistry, University of Colorado Boulder, Boulder, CO, 80309, USA.
[8]Department of Environmental Science, Policy, and Management, University of California Berkeley, Berkeley, CA, 94720, USA.
[9]Department of Civil and Environmental Engineering, University of California Berkeley, Berkeley, CA, 94720, USA.
[10]Department of Earth and Planetary Sciences, University of California Berkeley, Berkeley, CA, 94720, USA.

*Correspondence to:* Ronald C. Cohen (rccohen@berkeley.edu)

**Abstract.** Surface ozone concentrations are observed to increase with rising temperatures, but the mechanisms responsible for this effect in rural and remote continental regions remain uncertain. Better understanding of the effects of temperature on ozone is crucial to understanding global air quality and how it may be affected by climate change. We combine measurements from a focused ground campaign in summer 2013 with a long-term record from a forested site in the rural southeastern United States to examine how daily average temperature affects ozone production. We find that changes to local chemistry are key drivers of increased ozone concentrations on hotter days, with integrated daily ozone production increasing by 2.3 ppb $°C^{-1}$. Nearly half of this increase is attributable to temperature-driven increases in emissions of nitrogen oxides ($NO_x$), most likely by soil microbes. The increase of soil $NO_x$ emissions with temperature suggests that ozone will continue to increase with temperature in the future, even as direct anthropogenic $NO_x$ emissions decrease dramatically. The links between temperature, soil $NO_x$, and ozone form a positive climate feedback.

## 1 Introduction

Elevated concentrations of tropospheric ozone are an important contributor to anthropogenic radiative forcing and are associated with increased human mortality and decreased crop yields (Myhre et al., 2013; World Health Organization, 2005; Booker et al., 2009). Observations of increased surface ozone concentrations on hotter days are widely reported, but the mechanisms driving this relationship are poorly understood in regions and climates with low concentrations of nitrogen oxides

($NO_x \equiv NO + NO_2$). Understanding the mechanisms driving these increases is critical to effectively regulating ozone pollution and predicting the effects of global warming on air quality.

Several previous studies (e.g., Sillman and Samson, 1995; Weaver et al., 2009; Pusede et al., 2014) have used in situ observations and chemical transport models to examine the relationships between ozone and temperature. Typically observed slopes range from 1–6 ppb $°C^{-1}$, with greater values occurring in more polluted environments (Pusede et al., 2015). A few studies have also reported that this effect is nonlinear and can become significantly less strong at the highest temperatures (Steiner et al., 2010; Shen et al., 2016).

Increased ozone concentrations with temperature in urban areas can be well explained by increased ozone production caused by greater emissions of volatile organic compounds (VOCs) and decreased sequestration of $NO_x$ in short-term reservoirs (Jacob and Winner, 2009). In contrast, there is little consensus about the mechanisms responsible for temperature-dependent changes in ozone concentrations in rural and remote environments. Arguments in favor of large-scale changes in atmospheric circulation and in favor of local changes in the chemical production and loss of ozone have both been presented (Barnes and Fiore, 2013; Steiner et al., 2006). Regional stagnation episodes, often associated with elevated temperatures, allow ozone to accumulate over several days and are known to contribute significantly to the ozone-temperature relationship (Jacob et al., 1993). How various temperature-dependent chemical effects interact and their relative contributions to ozone production are not well understood outside of polluted environments.

Summer daytime ozone concentrations at rural sites in the United States typically range from 35–55 ppb (Cooper et al., 2012), sufficient to cause harm to humans, crops, and the climate. Epidemiological studies and meta-analyses investigating the relationship between ozone and daily mortality have found significant effects in small cities and rural locations, with some studies suggesting that increases in ozone may have a greater effect on daily mortality under less polluted conditions (Vedal et al., 2002; Ito et al., 2005; Atkinson et al., 2012). Studies of crop yield and plant health have traditionally used a threshold of 40 ppb when investigating the effects of ozone exposure, but many crops have been shown to experience reduced yields when exposed to ozone concentrations as low as 20 ppb (Pleijel et al., 2004; Booker et al., 2009). From a regulatory perspective, elevated regional background ozone can strongly exacerbate ozone pollution and the probability of regulatory exceedances in urban areas such as Houston (Berlin et al., 2013). Understanding the behavior of $O_3$ in the rural and remote areas that cover the majority of the land area of the Earth is therefore crucial for effectively predicting and controlling air quality now and in the future.

In this paper we use observations from Centreville, Alabama (CTR), a rural site in the southeastern United States (Fig. S1), to investigate how temperature affects ozone production. Long-term monitoring from the SouthEastern Aerosol Research and CHaracterization (SEARCH) network shows that ozone increases significantly with temperature at this site (Fig. 1), despite being in a low-$NO_x$ environment where the predicted response of the instantaneous ozone production rate to temperature is small (Pusede et al., 2015). We combine this record with extensive measurements from the Southern Oxidant and Aerosol Study (SOAS) in summer 2013 to explicitly calculate daily integrated ozone production and $NO_x$ loss as a function of daily average temperature. We find that changes in local chemistry are important drivers of the increase in ozone concentrations observed at this site, and that increased $NO_x$ emissions are responsible for 40% of the temperature-dependent increase in daily

integrated ozone production. We expect similar effects to be present in other low-$NO_x$ areas with high concentrations of VOCs, where the chemistry of alkyl and multifunctional nitrates is the majority pathway for permanent $NO_x$ loss.

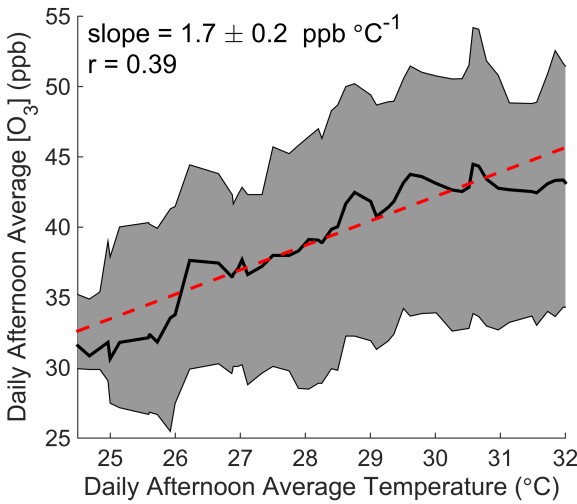

**Figure 1.** The $O_3$-temperature relationship in Centreville, Alabama. Daily afternoon (12 pm–4 pm) average ozone concentration is shown as a function of temperature from June–August 2010–2014 at the SEARCH CTR site. The black line and shaded gray region show the running median and interquartile range of ozone with temperature. The red line represents a fit to all daily data points.

## 2 Chemistry of ozone production and predicted response to temperature

Observed $O_3$-temperature relationships are caused by a combination of chemical changes to the production and loss of $O_3$
and changes to atmospheric circulation that determine advection and mixing. To begin separating these effects, we consider the chemical production of ozone ($PO_3$) and how it changes with temperature. Temperature-dependent changes in ozone production may be driven directly by temperature, or by another meteorological parameter that co-varies with temperature, such as solar radiation.

Ozone is produced in the troposphere when NO is converted to $NO_2$ by reaction with $HO_2$ or $RO_2$ in the linked $HO_x$ and
$NO_x$ cycles (Fig. 2a). $HO_2$ and $RO_2$ radicals are generated in the $HO_x$ cycle when a VOC reacts with OH in the presence of $NO_x$. In one turn of the cycle, the VOC is oxidized, OH is regenerated, and two molecules of $O_3$ are formed. The reactions that drive these catalytic cycles forward are in constant competition with reactions that remove radicals from the atmosphere, terminating the cycles. Termination can occur either through the association of two $HO_x$ radicals to form inorganic or organic peroxides, or through the association of $HO_x$ and $NO_x$ radicals to form nitric acid or an organic nitrate.
The balance between propagating and terminating reactions causes $PO_3$ to be a non-linear function of the $NO_x$ and VOC reactivity (VOCR), as well as the production rate of $HO_x$ radicals ($PHO_x$). The largest source of $HO_x$ radicals in the summertime is the photolysis of $O_3$ followed by reaction with water vapor to produce OH; additional sources include the photolysis of

formaldehyde and peroxides, ozonolysis of alkenes, and isomerization pathways in the oxidation of isoprene and other VOCs. To understand the response of ozone production to changes in chemistry, we use a simplified framework based on the balance of $HO_x$ radical production and loss (Farmer et al., 2011).

Under high or moderate $NO_x$ conditions, the primary loss process of $HO_2$ and $RO_2$ radicals is reaction with NO and the concentration of OH radicals can be expressed as a quadratic equation. To modify this approach to work under low-$NO_x$ conditions, reactions between $HO_x$ radicals must also be included, leading to a set of 4 algebraic equations that can be solved numerically (details given in Appendix A). Figure 2b shows the calculated rate of ozone production as a function of $NO_x$ at two different VOC reactivities. Depending on atmospheric conditions, the ozone production rate can either be $NO_x$-limited, where additional $NO_x$ causes $PO_3$ to increase, or $NO_x$-saturated, where additional $NO_x$ suppresses ozone formation.

When considering day-to-day variations, the total amount of ozone produced over the course of a day ($\int PO_3$) is a more representative metric than the instantaneous ozone production rate. Total daily ozone production depends on all of the factors that affect $PO_3$ as well as their diurnal evolution. In places where ozone production is $NO_x$-limited, changes to chemistry with temperature that affect the $NO_x$ loss rate ($\mathcal{L}NO_x$) can affect $\int PO_3$ by changing the amount of $NO_x$ available for photochemistry later in the day (Hirsch et al., 1996).

Permanent $NO_x$ loss occurs through two primary pathways in the troposphere: the association of OH and $NO_2$ to form $HNO_3$, and through the chemistry of alkyl and multifunctional nitrates ($\Sigma RONO_2$). These organic nitrates are formed as a minor channel of the $RO_2 + NO$ reaction, with the alkyl nitrate branching ratio $\alpha_i$ ranging from near zero for small hydrocarbons to over 0.20 for monoterpenes and long-chain alkanes (Perring et al., 2013). The overall alkyl nitrate branching ratio $\alpha_{eff}$ represents the reactivity-weighted average of $\alpha_i$ for all VOCs. While some fraction of $\Sigma RONO_2$ quickly recycles $NO_x$ to the atmosphere, a significant fraction $\eta$ permanently removes $NO_x$ through deposition and hydrolysis (e.g., Browne et al., 2013). Romer et al. (2016) determined that $\eta = 0.55$ during SOAS and was controlled primarily by the hydrolysis of isoprene hydroxy-nitrates. Because the hydrolysis rate is set primarily by the distribution of nitrate isomers, which does not change appreciably with temperature, we assume that $\eta$ is constant with temperature in this study (Hu et al., 2011; Peeters et al., 2014). Deposition is only a minor loss process for $\Sigma RONO_2$, therefore any changes in the deposition rate with temperature will have at most a minor effect on $\eta$.

$NO_x$ also has several temporary sinks that can sequester $NO_x$, most importantly peroxy acyl nitrate (PAN). In the summertime southeastern United States, the lifetime of PAN is typically 1–2 hours, too short to act as a permanent sink of $NO_x$. Past studies in forested regions have found remarkably little variation in PAN with temperature, due to compensating changes in both its production and loss (e.g., LaFranchi et al., 2009). As a result, the formation or destruction of PAN does not contribute significantly to net ozone production or $NO_x$ loss and we do not include it in these calculations.

The ozone production efficiency (OPE $\equiv PO_3/\mathcal{L}NO_x$) represents the number of ozone molecules formed per molecule of $NO_x$ consumed and directly links the ozone and $NO_x$ budgets. Because OPE accounts for changes in both $PO_3$ and $\mathcal{L}NO_x$, the temperature response of OPE captures feedbacks in ozone production chemistry that $PO_3$ alone does not.

As the concentration of $NO_x$ decreases and VOCR increases, the fraction of $NO_x$ loss that takes place via $HNO_3$ chemistry decreases and the OPE increases (Fig. 2c). The relative importance of $HNO_3$ and $RONO_2$ chemistry determines the relation-

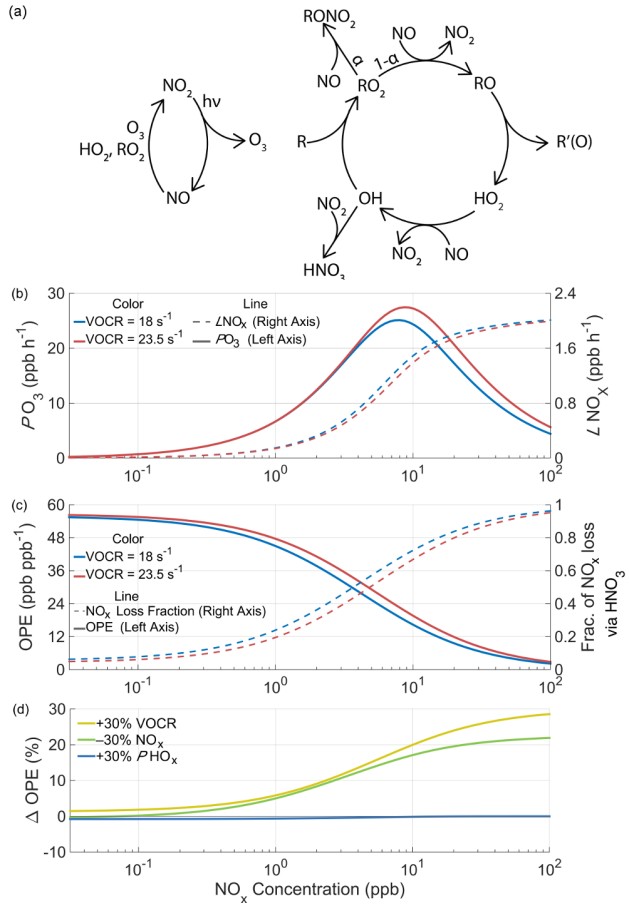

**Figure 2.** The chemistry of ozone production and $NO_x$ loss in the troposphere. (a) Schematic of the linked $NO_x$ and $HO_x$ cycles that lead to net ozone production. (b) The calculated instantaneous $O_3$ production rate and $NO_x$ loss rate as a function of $NO_x$ and VOCR, with fixed $PHO_x$, $\eta$, and $\alpha_{eff}$. (c) OPE and the fraction of $NO_x$ loss that takes place via $HNO_3$ chemistry under the same conditions as (b). (d) The percent change in ozone production efficiency caused by chemical changes as a function of $NO_x$.

ship between $PO_3$ and $\mathcal{L}NO_x$. When $HNO_3$ is the most important $NO_x$ loss pathway, $O_3$ production and $NO_x$ loss occur through separate channels. $O_3$ production occurs when OH reacts with a VOC, generating $RO_2$ and $HO_2$ radicals; $NO_x$ loss primarily occurs when OH reacts with $NO_2$. Although these channels are linked by a shared dependence on OH, the relative importance of these pathways can vary. For example, under these conditions an increase in VOCR will cause $NO_x$ loss to
5   decrease, ozone production to increase, and OPE to increase (Fig. 2b–c).

    In contrast, when $RONO_2$ chemistry dominates $NO_x$ loss, ozone production and $NO_x$ loss are intrinsically linked by their shared dependence on the $RO_2 + NO$ reaction. This reaction produces $O_3$ in its main channel and consumes $NO_x$ in the minor channel that forms organic nitrates, with the ratio between these two channels set by $\alpha_{eff}$. Under these conditions, changes to the chemistry that do not affect $\alpha_{eff}$ have a minimal effect on OPE (Fig. 2d) and the OPE can be considered to be unvarying with

temperature. An increase in VOCR or a decrease in $NO_x$ will affect both $NO_x$ loss and ozone production equally, because both processes are dependent on the same set of reactions. Because of this change in behavior, from variable OPE to fixed OPE, the drivers of the $O_3$-T relationship are expected to be categorically different in areas where $RONO_2$ chemistry dominates $NO_x$ loss. As a result, the effects that cause $O_3$ to increase with temperature in urban and other polluted regions, where $HNO_3$ chemistry dominates $NO_x$ loss, are unlikely to apply in areas with low concentrations of $NO_x$ and high concentrations of reactive VOCs, where $RONO_2$ chemistry is most important. In these areas, more $NO_x$ must be oxidized in order to produce more $O_3$.

## 3 Observed response of ozone production to temperature

### 3.1 Measurements during SOAS

The theoretical results presented in Fig. 2 can be compared to the observed behavior during SOAS. Measurements during SOAS have been described in detail elsewhere (e.g., Hidy et al., 2014; Romer et al., 2016; Feiner et al., 2016) and are summarized below. The primary ground site for SOAS was co-located with the CTR site of the SEARCH network (32.90289° N, 87.24968° W), in a clearing surrounded by a dense mixed forest (Hansen et al., 2003). Direct anthropogenic emissions of $NO_x$ near this site are estimated to be low and predominantly from mobile sources (Hidy et al., 2014). Figure S1 shows the location of the CTR site relative to major population centers in the region. Measurements taken as part of the SEARCH network were located on a 10 m tower approximately 100 m away from the forest edge, while the other measurements from the SOAS campaign used in this analysis were located on a 20 m walk-up tower at the edge of the forest. Species measured on both the SOAS walk-up tower and the SEARCH platform were well correlated with each other, indicating that similar airmasses were sampled at both locations.

Several chemical and meteorological measurements used in this study, including $NO_x$, $O_3$, total reactive nitrogen ($NO_y$), and temperature, were collected by Atmospheric Research and Analysis (ARA) as part of SEARCH (Hidy et al., 2014). NO was measured using the chemiluminescent reaction of NO with excess ozone. $NO_2$ was measured based on the same principle, using blue LED photolysis to convert $NO_2$ to NO. The photolytic conversion of $NO_2$ to NO is nearly 100% efficient and does not affect higher oxides of nitrogen (Ryerson et al., 2000). Ozone was measured using a commercially available ozone analyzer (Thermo-Scientific 49i).

During the SOAS campaign, $NO_2$, total peroxy nitrates ($\Sigma PNs$), and total alkyl and multifunctional nitrates ($\Sigma RONO_2$) were measured via thermal dissociation laser-induced fluorescence, as described by Day et al. (2002). An NO chemiluminescence instrument located on the walk-up tower provided additional measurements of NO co-located with the other SOAS measurements (Min et al., 2014).

$HO_x$ radicals were measured with the Penn State Ground-based Tropospheric Hydrogen Oxides Sensor (GTHOS), which uses laser induced fluorescence to measure OH (Faloona et al., 2004). $HO_2$ was also measured in this instrument by adding NO to convert $HO_2$ to OH. $C_3F_6$ was periodically added to the sampling inlet to quantify the interference from internally generated OH (Feiner et al., 2016). Measurements of total OH reactivity (OHR $\equiv$ inverse OH lifetime) were made by sampling ambient

air, injecting OH, and letting the mixture react for a variable period of time. The slope of the OH signal versus reaction time provides a top-down measure of OHR (Mao et al., 2009).

A wide range of VOCs were measured during SOAS using gas chromatography-mass spectrometry (GC-MS). Samples were collected in a liquid-nitrogen cooled trap for five minutes, then transferred by heating onto an analytical column, and detected using an electron-impact quadrupole mass-spectrometer (Gilman et al., 2010). This system is able to quantify a wide range of compounds including alkanes, alkenes, aromatics, isoprene, and multiple monoterpenes at a time resolution of 30 minutes. Methyl vinyl ketone (MVK) and methacrolein (MACR) were measured individually by GC-MS and their sum was also measured using a proton transfer reaction mass spectrometer (PTR-MS) (Kaser et al., 2013). The calculated rates of ozone production and $NO_x$ loss do not change significantly depending on which measurement is used.

## 3.2 Calculation of $\int PO_3$ and effects of temperature

During the SOAS campaign, afternoon concentrations of $NO_x$ averaged 0.3 ppb and concentrations of isoprene 5.5 ppb (Fig. S2). $\Sigma RONO_2$ chemistry was responsible for over three-quarters of the permanent $NO_x$ loss (Romer et al., 2016). Daily average afternoon (12 pm–4 pm) ozone concentrations increased with daily average afternoon temperature during SOAS ($2.3 \pm 1$ ppb $°C^{-1}$). This trend is greater than the long-term trend reported by the SEARCH network, but the difference is not statistically significant.

Measurements of NO, $NO_2$, OH, $HO_2$, and a wide range of VOCs (Table S1) were used to calculate the steady state concentrations of $RO_2$ radicals using the Master Chemical Mechanism v3.3.1, run in a MATLAB framework (Jenkin et al., 2015; Wolfe et al., 2016). Before 24 June, $HO_2$ measurements are not available and steady state concentrations of both $HO_2$ and $RO_2$ were calculated. Input species were taken to 30 minute averages, and the model was run until radical concentrations reached steady state. Top-down measurements of OHR were used to include the contribution to ozone production from unmeasured VOCs.

To understand the day-to-day variation of ozone chemistry, the calculated ozone production rate was integrated from 6 am to 4 pm for each of the 24 days during the campaign period with greater than 75% data coverage of all input species. When plotted against daily average afternoon temperature, $\int PO_3$ is seen to increase strongly with temperature ($2.3 \pm 0.6$ ppb $°C^{-1}$, Fig. 3a). The change in $\int PO_3$ with temperature demonstrates that local chemistry is an important contributor to the observed $O_3$-T relationship; however, the observed $O_3$-T trend also includes the effects of chemical loss, advection, entrainment, and multi-day buildup on overall $O_3$ concentration (e.g., Baumann et al., 2000).

While elevated temperatures are associated with enhanced production of ozone, they are also associated with increased chemical loss. The chemical loss of ozone occurs through three main pathways in this region: photolysis followed by reaction with $H_2O$, reaction with $HO_2$, and reaction with VOCs (Frost et al., 1998). The loss of $O_3$ was calculated for each of these pathways, and then integrated over the course of the day to determine total daily ozone loss ($\int \mathcal{L} O_3$). Chemical loss of ozone is found to increase with temperature ($1.1 \pm 0.3$ ppb $°C^{-1}$, Fig. 3b), but much less than the chemical production.

The difference between the trend in the net chemical production and loss of $O_3$ and the trend in ozone concentration gives a rough estimate of how non-chemical processes contribute to the ozone-temperature relationship. We calculate that non-

chemical processes cause $O_3$ to increase by $1\pm1.2$ ppb $°C^{-1}$. This approach does not take into account the interactions between chemical and non-chemical effects, such as how changes to advection and mixing may impact concentrations of VOCs, $NO_x$, and other reactants. Although the large uncertainty does not allow for quantitative analysis, qualitatively, chemical and non-chemical processes are both found to be important contributors to the ozone-temperature relationship. Other approaches, such as chemical transport models, that can more directly investigate and control specific physical processes are likely to be better suited to calculating the contribution of non-chemical processes to the ozone-temperature relationship (e.g., Fu et al., 2015).

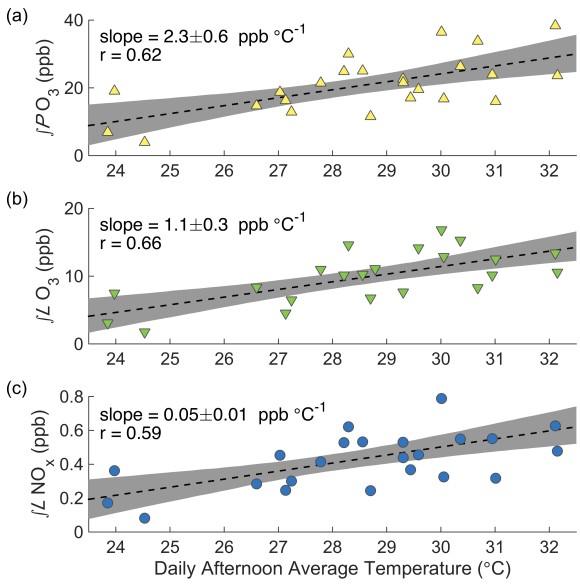

**Figure 3.** Observed dependence of daily $\int PO_3$ (a), $\int \mathcal{L}O_3$ (b), and $\int \mathcal{L}NO_x$ (c) on daily afternoon average temperature during SOAS. Each point shows the afternoon average temperature and integrated production or loss for a single day. Black lines show a least squares fit to all points; shaded areas show the 90% confidence limits of the fit calculated via bootstrap sampling.

Using the same calculated radical concentrations, the rate of $NO_x$ loss was calculated as the rate of direct $HNO_3$ production plus the fraction $\eta$ of alkyl nitrate production that leads to permanent $NO_x$ loss. Figure 3c shows the increase in $\int \mathcal{L}NO_x$ with temperature for the SOAS campaign ($0.05 \pm 0.01$ ppb $°C^{-1}$). As expected from the importance of $RONO_2$ chemistry to $NO_x$ loss, $\int \mathcal{L}NO_x$ and $\int PO_3$ are tightly correlated ($r^2 = 0.90$), and OPE is high (OPE average $45 \pm 3$ ppb ppb$^{-1}$) and is effectively constant with temperature (calculated trend $0.2 \pm 0.6$ $°C^{-1}$). Therefore, the increase in $\int PO_3$ with temperature is not caused by more efficient production of ozone while the same amount of $NO_x$ is consumed.

OPE can also be estimated from the ratio of odd oxygen ($O_x \equiv O_3 + NO_2$) to $NO_x$ oxidation products ($NO_z \equiv NO_y - NO_x$) (Trainer et al., 1993). The afternoon ratio of $O_x$ to $NO_z$ during SOAS varied from 43–67 (interquartile range), slightly higher than the average ratio of $\int PO_3$ to $\int \mathcal{L}NO_x$. However, since the $O_x$ to $NO_z$ ratio includes the effects of chemical loss and

transport, which the ratio of $\int PO_3$ to $\int \mathcal{L}NO_x$ does not, these two values are not expected to be equivalent, particularly in non-polluted areas.

The trend in $\int PO_3$ with temperature is robust and extends beyond the short temporal window of the SOAS campaign. Although long-term measurements of $HO_x$ and VOCs are not available, the ozone production rate can be estimated from SEARCH measurements using the deviation of NO and $NO_2$ from photostationary state (Eq. 1) (Baumann et al., 2000; Pusede et al., 2015).

$$PO_3 = j_{NO_2}[NO_2] - k_{NO+O_3}[NO][O_3] \tag{1}$$

The $NO_2$ photolysis rate was parameterized as a quadratic function of total solar radiation (Trebs et al., 2009). Using this method and scaling the result to match the values calculated using steady-state $RO_2$ concentrations during SOAS, we find that $\int PO_3$ increased by $2.3 \pm 0.8$ ppb $°C^{-1}$ during June–August 2010–2014 (Fig. S3). Without scaling, the long-term trend in $\int PO_3$ with temperature is $4.0 \pm 0.5$ ppb $°C^{-1}$. Based on the long-term SEARCH record, we do not find evidence that the relationship between ozone concentration or ozone production changes significantly at the highest temperatures (the top 5% of observations). This agrees broadly with Shen et al. (2016), who found that ozone suppression at extreme temperatures to be uncommon in the southeastern United States.

## 4 Drivers of increased ozone production

While the increase in ozone production is accompanied by an observed increase in ozone concentration, the increase in $NO_x$ loss is not accompanied by a significant decrease in $NO_x$ concentration ($-0.002 \pm 0.01$ ppb $°C^{-1}$, Fig. 4a). For this to occur, $NO_x$ must have a source that increases with temperature to compensate for its increased loss. One possible explanation is that the increased thermal decomposition rate of peroxy nitrates ($\Sigma PNs$) causes less $NO_x$ to be sequestered in these short-term reservoirs. This is not the case during SOAS. The increased decomposition rate of peroxy nitrates is counteracted by an increase in their production rate, such that the average concentration of total peroxy nitrates shows no decrease with temperature (Fig. 4b).

More generally, increased transformations from $NO_x$ oxidation products back into $NO_x$ cannot explain the observations. The concentration of $NO_y$ increases significantly with temperature (Fig. 4c). Because $NO_y$ includes $NO_x$ as well as all of its reservoirs and sinks, changes in the transformation rates between $NO_x$ and its oxidation products cannot explain the increase of $NO_y$ with temperature. There must be a source of $NO_y$, not just of $NO_x$, that increases with temperature.

Data from the SEARCH network indicate that the increase in $NO_y$ with temperature observed during SOAS is primarily a local effect. Measurements from June–August 2010–2014 show a consistent increase of $NO_y$ with temperature at the two rural monitoring sites in the network, but total $NO_y$ decreases with temperature at the four urban and suburban sites (Table S2). The increase in $NO_y$ with temperature therefore cannot be explained by regional meteorological effects, since those would lead to similar relationships between $NO_y$ and temperature across the southeastern United States.

Measurements at night and in the early morning, before significant photochemistry has occurred, show a strong temperature-dependent increase of $NO_x$ over the course of the night. Because surface wind speeds are low at night and the increase in $NO_x$

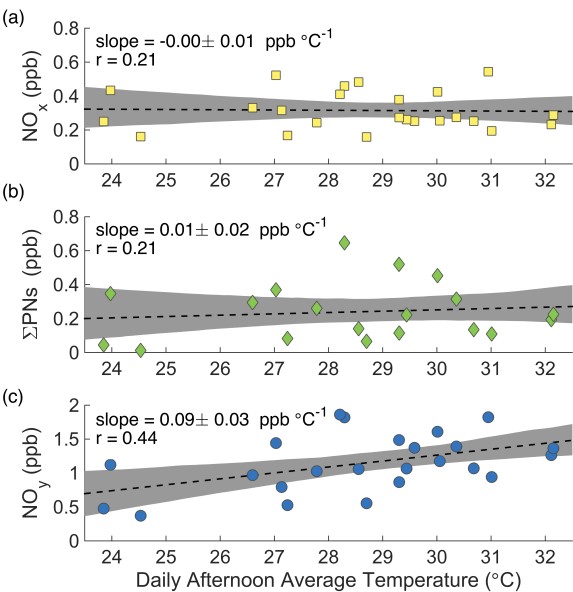

**Figure 4.** Afternoon average concentrations of $NO_x$ (a), $\Sigma PNs$ (b), and $NO_y$ (c) at the CTR site as a function of daily average afternoon temperature during SOAS.

at night is not accompanied by large increases in $NO_x$ oxidation products, the increase in $NO_x$ must be caused by emissions local to the CTR site.

The consistent increase of $NO_x$ over the course of the night can be used to quantitatively measure the local $NO_x$ emissions rate. Figure 5 shows the temperature-dependent increase of $NO_x$ relative to the concentration of $NO_x$ at 4 pm the day before, separating the effects of the previous day from the nighttime increase. Measurements from June–August 2010–2014 from the CTR SEARCH network site are used to obtain more representative statistics. The average rate of $NO_x$ increase during the night is 0.095 ppb $h^{-1}$. To account for the chemical removal of $NO_x$, the cumulative loss of $NO_x$ during the night was added to the observations. During SOAS, the nighttime loss of $NO_x$ occurred almost exclusively through the reaction of $NO_2$ with $O_3$ to form $NO_3$, which then reacted with a VOC to form an organic nitrate (Ayres et al., 2015). $N_2O_5$ chemistry made a negligible contribution to total $NO_x$ loss. The loss rate of $NO_x$ during the night was therefore calculated as the rate of reaction of $NO_2$ with $O_3$. In this form, the rate of increase of the adjusted $NO_x$ concentrations ($NO_x^*$) is equal to the local $NO_x$ emission rate. The emission rate of $NO_x$ and its temperature dependence were calculated by a linear regression following the form of Eq. (2), where the adjusted concentration of $NO_x$ depends both on time (H = hours after 4 pm) and temperature (T).

$$NO_x^* = (\alpha T + \beta)H + b \tag{2}$$

In this regression, the fitted parameter $\alpha$ represents the increase of $NO_x$ emissions with temperature and the average value of $\alpha T + \beta$ provides an estimated $NO_x$ emission rate.

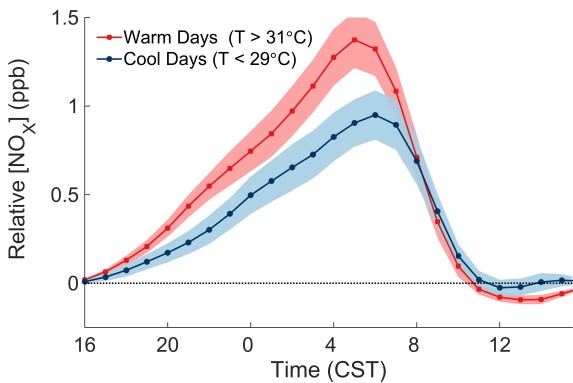

**Figure 5.** Concentrations of $NO_x$ relative to their concentration at 4 pm the day before over June–August 2010–2014 at the CTR site. The thick lines and shaded areas show the hourly mean and 90% confidence interval of the mean for cooler and warmer days.

Because emissions are localized to the surface, the effective depth of the nighttime boundary layer must also be accounted for, which we estimate to be 150 m. This agrees well with the derived mixing heights from daily 5 am sonde launches at the Birmingham (BHM) airport (Durre and Yin, 2008) and past estimates of the nocturnal boundary layer height (e.g., Liu and Liang, 2010; VandenBoer et al., 2013), while it is significantly lower than the average ceilometer-reported 5 am boundary layer height of 400 m during SOAS (Fig. S4).

After accounting for these factors, the $NO_x$ emissions rate is calculated to be 7.4 ppt m s$^{-1}$ or 4.2 ng N m$^{-2}$ s$^{-1}$. Based on the change in slope with temperature, the emissions rate is estimated to increase by 0.4 ppt m s$^{-1}$ °C$^{-1}$. The rise in $NO_x$ emissions with temperature over 24 hours agrees to within the uncertainty with the increase of daily $\int \mathcal{L}NO_x$ with temperature, sufficient to explain why afternoon $NO_x$ concentrations are not observed to decrease with temperature even as their loss rate increases.

The inferred local $NO_x$ source bears all the hallmarks of soil microbial emissions ($S_{NO_x}$). Soil microbes emit $NO_x$ as a byproduct of both nitrification and denitrification, and the rate of $NO_x$ emissions correlates strongly with microbial activity in soil (Pilegaard, 2013). The inferred $NO_x$ source is active during day and night, increases strongly with temperature, and is present in a rural area with low anthropogenic emissions. The only plausible source of $NO_x$ that matches all of these constraints is soil microbial emissions near to the SOAS site. Soil $NO_x$ emissions also depend on the water content and nitrogen availability, neither of which is generally limiting in the southeastern United States (e.g., Hickman et al., 2010). The most likely anthropogenic sources of $NO_x$ at this location are mobile sources, which are not thought to change significantly with temperature (Singh and Sloan, 2006) and therefore cannot explain the results of Fig. 5.

To calculate how the increase in $NO_x$ emissions affects ozone production, we use the same chemical framework from Fig. 2. For each half-hour period the average value of the input parameters and their temperature dependence during the SOAS campaign were calculated (Fig. S5). The diurnal cycle and trend with temperature of all model inputs were then used to calculate total daily ozone production as a function of temperature (Fig. S6). By altering whether the temperature dependence

for each parameter is included, the overall trend in $\int PO_3$ can be decomposed into individual components (Fig. 6). The effect of increased $NO_x$ emissions was calculated by fixing the trend in $NO_x$ with temperature to match the trend in $\int \mathcal{L}NO_x$. We find that the increase of $NO_x$ emissions with temperature accounts for 40% of the increase in $\int PO_3$ with temperature, or approximately 0.9 ppb $°C^{-1}$. The other 60% is caused primarily by the increase of $PHO_x$ with temperature. The increase in $PHO_x$ with temperature is most likely caused by changes in solar radiation, which is well correlated with the total $PHO_x$ rate (Fig. S7a) and increases strongly with temperature. In contrast, water vapor is not correlated with total $PHO_x$ (Fig. S7b). Although VOCR increases strongly with temperature, the $RONO_2$-dominated $NO_x$ chemistry causes neither the ozone production rate nor the $NO_x$ loss rate to be sensitive to this increase, leading to the minimal effect of VOCR on $\int PO_3$.

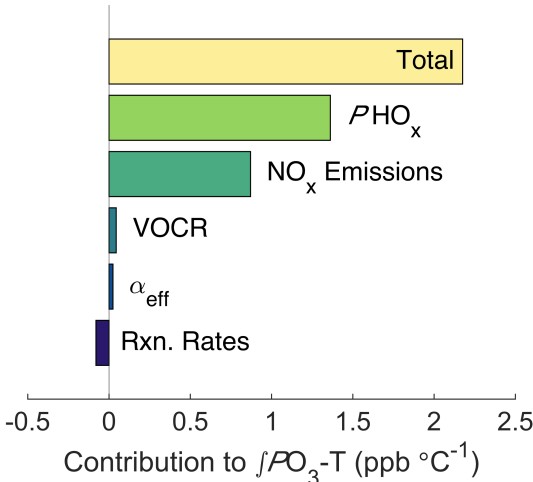

**Figure 6.** Decomposed effects of ozone and temperature. The top bar shows the model-calculated $\int PO_3$-T trend, all other bars show how the $\int PO_3$-T slope changes when the temperature dependence of each factor is removed.

## 5 Conclusions

Changes in $NO_x$ emissions with temperature have an outsized effect when considering the impacts of ozone on human health and climate. At the CTR site and other areas where OPE does not vary with temperature, the total amount of ozone produced on weekly or monthly timescales is directly proportional to the amount of available $NO_x$. While faster oxidation on hotter days causes more ozone to be produced, without changes in $NO_x$ emissions there would be an associated decrease in ozone production on subsequent days, because the $NO_x$ necessary for ozone production would be depleted. In contrast, increased $NO_x$ emissions can cause weekly or monthly average ozone concentrations to increase with temperature. Change in long-term average ozone concentrations is often more important to the ozone climate feedback and human health than day-to-day variation. The mechanisms described here are likely to be active in all areas with low concentrations of $NO_x$ and high

concentrations of reactive VOCs. Only regions where $RONO_2$ chemistry is the dominant pathway for $NO_x$ loss have effectively constant OPE with temperature, but the effect of soil $NO_x$ emissions on ozone production is widespread.

Past direct measurements of soil $NO_x$ using soil chambers have found enormous variability, both between sites and within different plots in the same field. Pilegaard et al. (2006) found variability of a factor of over 100 between soil $NO_x$ emissions in different European forests. Within the southeastern United States, direct measurements at forested sites have reported emissions rates ranging from 0.1–10 ng N $m^{-2}$ $s^{-1}$ (Williams and Fehsenfeld, 1991; Thornton et al., 1997; Hickman et al., 2010). Besides temperature, the most important variables affecting soil $NO_x$ emissions are typically nitrogen availability and soil water content, as well as plant cover and soil pH (Pilegaard, 2013). In very wet environments, soil microbes typically emit $N_2O$ or $N_2$ instead of $NO_x$, and in arid environments soil emissions of HONO can be equal to or larger than soil $NO_x$ emissions (Oswald et al., 2013). Although conditions at the CTR site are too wet and acidic for soil HONO emissions to be significant, in environments where soil HONO emissions are large, they would likely have an even greater effect on ozone production by acting as a source of both $NO_x$ and $HO_x$ radicals.

The variability between sites and the interaction between several biotic and abiotic factors make it difficult to apply regional or model estimates of soil $NO_x$ emissions to a particular location. Our approach from this study, using observations of the nighttime atmosphere to determine the $NO_x$ emissions rate, helps span the gap between soil chambers and the regional atmosphere. Although soil $NO_x$ emissions depend on several environmental factors, process-driven models predict that the response of soil $NO_x$ emissions to global warming will be driven primarily by the increase in temperature (Kesik et al., 2006).

While soil $NO_x$ emissions have been known and studied for decades, the impacts of soil $NO_x$ emissions on ozone from non-agricultural regions was often found to be insignificant compared to anthropogenic sources (e.g., Davidson et al., 1998). Years of declining anthropogenic $NO_x$ emissions in the United States and recent higher estimates for forest soil $NO_x$ emissions (e.g., Hickman et al., 2010) mean that this is no longer the case. Non-agricultural soil $NO_x$ emissions may now account for nearly a third of total $NO_x$ emissions in the summertime southeastern United States (Travis et al., 2016), and have significant effects on regional ozone production.

The rise in ozone production caused by increased $NO_x$ emissions on hotter days established here suggests that the relationship between ozone and temperature will be positive under a wider range of conditions than previously thought. This includes 1. the pre-industrial atmosphere, 2. present day rural continental locations, and 3. future scenarios with dramatically reduced anthropogenic $NO_x$ emissions.

1. In pre-industrial times, semi-quantitative measurements of ozone show significantly lower concentrations of ozone than currently observed in rural and remote regions or generally predicted by global models (Cooper et al., 2014). While alkyl nitrate chemistry establishes an upper limit to the ozone production efficiency under low-$NO_x$ conditions, the significant contribution of $S_{NO_x}$ to ozone production makes reconciling the semi-quantitative measurements with model predictions more difficult and suggests that natural emissions of $NO_x$ in pre-industrial models may be over-estimated (Mickley et al., 2001).

2. In the present day, effective ozone regulation, especially on hot days, requires taking into account the effect of $S_{NO_x}$. Because these emissions are distributed over broad areas and are not directly anthropogenic, they present additional challenges to air quality management. Indirect approaches, such as changes to fertilizer application practices, have the potential to significantly reduce $S_{NO_x}$ from agricultural regions (Oikawa et al., 2015). Decreases in direct anthropogenic $NO_x$ emissions may also lead to a decrease in $S_{NO_x}$ by decreasing the amount of nitrogen available to the ecosystem (Pilegaard, 2013).

3. In the future, because soil $NO_x$ emissions lead to the formation of ozone, itself an important greenhouse gas, the increase of soil $NO_x$ emissions with temperature represents a positive climate feedback and an additional link between changes to the nitrogen cycle and the environment. The effects of increased ozone pollution to plants, including reduced photosynthesis and slower growth, have the potential to alter the carbon cycle on a regional scale (Heagle, 1989; Booker et al., 2009). Soil $NO_x$ emissions therefore represent an additional link between the nitrogen and carbon cycles that should be included when considering the consequences of a warming world.

*Data availability.* Measurements from the SOAS campaign are available at https://esrl.noaa.gov/csd/projects/senex/.

**Appendix A**

**A1 Analytical $PO_3$ model**

To conceptually understand $O_3$ production and $NO_x$ loss, we use a simplified framework similar to that described by Farmer et al. (2011). This framework uses fixed values of total organic reactivity (VOCR), alkyl nitrate branching ratio $\alpha$ and loss efficiency $\eta$, $NO_x$, and $HO_x$ radical production rate ($PHO_x$).

Since $HO_x$ radicals are highly reactive, it is a valid assumption under nearly all $NO_x$ concentrations that $HO_x$ radicals are in steady-state and that $PHO_x$ is equal to the gross $HO_x$ loss rate (Eq. A1).

$$
\begin{aligned}
PHO_x = {} & k_{OH+NO_2}[OH][NO_2] + \alpha \cdot k_{RO_2+NO}[RO_2][NO] + 2k_{HO_2+HO_2}[HO_2][HO_2] \\
& + 2k_{RO_2+HO_2}[RO_2][HO_2] + 2k_{RO_2+RO_2}[RO_2][RO_2]
\end{aligned} \tag{A1}
$$

Individual $HO_x$ radicals (OH, $HO_2$, and $RO_2$) can also be assumed to be in steady state, such that their production and loss are equal. Under low-$NO_x$ conditions, the reactions that initiate and terminate the $HO_x$ cycle must be included as well as the cycling rate. We further constrain the model by requiring that the concentration of $HO_2$ and $RO_2$ radicals be equal. This constraint is satisfied by introducing an additional parameter $c$ which allows $PHO_x$ to produce both $HO_2$ and OH radicals in a varying ratio. These constraints provide a system of 4 equations that can be solved numerically (Eq. A2–A5).

$$
[OH] = \frac{k_{HO_2+NO}[HO_2][NO] + c \cdot PHO_x}{VOCR + k_{OH+NO_2}[NO_2]} \tag{A2}
$$

$$[RO_2] = \frac{[OH] \cdot VOCR}{k_{RO_2+NO}[NO] + k_{RO_2+HO_2}[HO_2] + 2k_{RO_2+RO_2}[RO_2]} \tag{A3}$$

$$[HO_2] = \frac{(1-\alpha)k_{RO_2+NO}[RO_2][NO] + (1-c)PHO_x}{k_{HO_2+NO}[NO] + 2k_{HO_2+HO_2}[HO_2] + k_{HO_2+RO_2}[RO_2]} \tag{A4}$$

$$[HO_2] = [RO_2] \tag{A5}$$

For the calculations in Fig. 2, the values of VOCR, $\alpha$, and $PHO_x$ were fixed at $18\,\mathrm{s}^{-1}$, 0.06, and $1.15 \times 10^7$ molec. $\mathrm{cm}^{-3}\,\mathrm{s}^{-1}$. Rate constants are taken from the IUPAC chemical kinetics database, assuming that all $RO_2$ radicals react with the kinetics of $CH_3CH_2O_2$ (Atkinson et al., 2006). The system of equations was solved numerically using the vpasolve function in MATLAB, subject to the constraints that $[OH], [HO_2]$, and $[RO_2]$ are positive and $c$ is between 0 and 1.

The resulting concentrations of $HO_x$ radicals can be used to calculate the rates of ozone production and $NO_x$ loss using Eq. (A6)–(A9).

$$PO_3 = (1-\alpha)k_{RO_2+NO}[RO_2][NO] + k_{HO_2+NO}[HO_2][NO] \tag{A6}$$

$$PHNO_3 = k_{OH+NO_2}[OH][NO_2] \tag{A7}$$

$$P\Sigma RONO_2 = \alpha \cdot k_{RO_2+NO}[RO_2][NO] \tag{A8}$$

$$LNO_x = PHNO_3 + \eta \cdot P\Sigma RONO_2 \tag{A9}$$

## A2   Decomposition of the $O_3$-Temperature Relationship

The simplified $HO_x$ model described above was used to decompose the contribution of different parameters to the increase of $\int PO_3$ with temperature. Peroxy nitrates are not included in this model, but because there is no significant trend in $\Sigma PNs$ with temperature their absence does not affect the results. To validate that this model gave accurate $\int PO_3$ results, it was first run using inputs based on measured values for each half-hour period:

- Model inputs of $NO_x$ were taken directly from measurements of $NO$ and $NO_2$

- VOCR was calculated as the measured OHR minus the reactivity of species that do not form $RO_2$ radicals (e.g., CO, $NO_2$)

- $PHO_x$ was calculated as equal to the measured rate of $HO_x$ loss, using Eq. (A1) and measured $HO_x$ radical concentrations

- $\alpha_{eff}$ was calculated as the reactivity-weighted average of $\alpha_i$ for all measured VOCs.

The comparison of $\int PO_3$ calculated from the full data set and that from the steady-state $HO_x$ model is shown in Fig. S6a.
The two calculations are well-correlated with a slope close to one, showing that the steady-state $HO_x$ model can accurately reproduce ozone production at this location.

To use this model to explore how ozone production changes with temperature, the diurnal cycle and trend in temperature of each of these inputs was calculated. Because the response to temperature is different at different times of day, the trend with temperature was calculated independently for each half-hour bin, and is shown in Fig. S5. These trends were used to construct
temperature-dependent diurnal cycles of each of the parameters, which were then used as inputs to the model at a range of daily average afternoon temperatures from 24–32 °C. Figure S6b shows that $\int PO_3$ calculated this way has a very similar trend with temperature as that using the full data set, although it cannot capture day-to-day variability not caused by temperature. The nonlinear shape of the trend with temperature is caused primarily by the imposed exponential increase of $PHO_x$ with temperature. Using a linear or quadratic increase of $PHO_x$ with temperature changes the shape of the increase but does not
significantly affect the overall $\int PO_3$-T slope.

*Competing interests.* The authors declare that no competing interests are present.

*Acknowledgements.* Financial and logistical support for SOAS was provided by the NSF, the Earth Observing Laboratory at the National Center for Atmospheric Research (operated by NSF), the personnel at Atmospheric Research and Analysis, and the Electric Power Research Institute. We are grateful to K. Olson and L. Zhang for assistance with measurements during SOAS. Funding for the SEARCH network was
provided by Southern Company Services and the Electric Power Research Institute. The Berkeley authors acknowledge the support of the NOAA Office of Global Programs grant NA13OAR4310067 and NSF grant AGS-1352972.

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
