# Peer review of "Effects of temperature-dependent $\mathbf{NO}_{\mathbf{x}}$ emissions on continental ozone production"

_Atmospheric Chemistry and Physics, 2017_

## Referee Comment (RC1) · Anonymous Referee #1 · 19 Oct 2017

This manuscript is well-written, within the scope of ACP, and provides valuable evidence for the increasing importance of emissions of NO$_x$ from soils to ozone production as temperatures increase. This manuscript should be published after minor revisions detailed below.

General Comments

1. The finding of increased soil NO$_x$ emission with temperature is valuable, and could be strengthened by a discussion of any known limitations on this effect, such as soil moisture or nitrogen availability. The authors discuss this briefly on page 12, but a more thorough discussion of what is known about microbes would be a valuable addition to

this manuscript.

2. The authors should improve the discussion of the effects of local meteorology on surface ozone. Jacob and Winner (2009) also discusses the strong positive relationship of ozone with temperature due to the association of temperature with regional stagnation. Was there stagnation on warmer days that would be a contributing factor to the ozone-temperature relationship? Convince the reader of the extent of the effect of the increased $NO_x$ emissions on this relationship in the context of likely different meteorology on hot days.

3. A final valuable addition would be a statement about whether the authors observe any breakdown of the observed ozone-T relationship at the highest temperatures, as found by Shen et al, 2016 (GRL), or whether their approach could be applied to this problem as well or would be impacted by this phenonemon.

Specific Comments

Page 2, line 17-18 – Could you clarify the point of Berlin et al, 2013? They are talking about 'background' ozone coming in to Houston, and I don't see the connection between your point about rural ozone and this paper.

Page 4, line 23-24 – You say, "When $HNO_3$ is the most important $NO_x$ loss pathway, $O_3$ production and $NO_x$ loss occur through separate channels and can change independently." Can you clarify this? Aren't both pathways competing for $NO_2$, so they are not actually independent? In the example that follows, more explicit statements of what is happening would be useful.

Page 4, line 30 – It is unclear to me whether you include thermal decomposition of PAN for example here, so that if temperature goes up, the effective yield of the sink would go down and OPE would not be fixed. Also, if you are integrating over a day, do you think that ignoring deposition is at all important?

Page 7, line 18 – How does an average OPE of 45 compare to OPE calculated from your model of $PO_3/LNO_x$?

Page 7, line 19 – Why do you say there is no OPE trend, but then provide a value (0.2)? If it is not statistically significant, don't show a number.

Page 10, line 20 – You say, "The increase of $PHO_x$ is mostly driven by increased solar radiation, and not by temperature directly." Could it not also be driven by increased water vapor with higher temperatures?

Page 12, line 22 – You say "These emissions cannot be regulated or controlled directly, and therefore present challenges to traditional air quality management techniques." Then this statement seems to be a contradiction - "Alternative approaches, such as changes to fertilizer application practices, have the potential to significantly reduce SNOx from agricultural regions (Oikawa et al., 2015)."

---

## Referee Comment (RC2) · Anonymous Referee #2 · 30 Nov 2017

Romer et al. disentangles the impact of different processes affecting the O3-T relationship in South Eastern US. The hypothesis and the arguments in the manuscript are well presented and provide robust evidence of the importance of soil-NOx for continental O3 production. Discussion of the results and their implications is scientifically sound. The manuscript should be published in ACP. I only have two minor comments that I would like the authors to address.

Minor comments

1. At page 9 lines 3-4 the loss of NOx due to NO2 + O3 reaction is taken into account to extract the increase in NOx due to soil emissions. I wonder how much of a change

would accounting for the NO2 + NO3 reaction which has a five order of magnitude higher rate constant. I expect no NO3 measurements for the CTR SEARCH network but for the SOAS measurements (Ayres et al. 2015) it should be possible.

2. The authors are only concerned with soil-NOx emissions although it is now known that soil bacteria are a comparable source of HONO (Oswald et al. 2013). HONO was measured during SOAS (https://data.eol.ucar.edu/dataset/373.037) and its impact on PO3-T is likely convoluted in the 60% contribution of PHOx shown in Fig. 6. In the manuscript it is stated that PHOx is mainly driven by increased solar radiation without showing (or explicitly pointing to) relevant data. However, soil-HONO emissions might also contribute to the PHOx category in Fig. 6. Could the authors attempt a sensitivity analysis or at least discussion of the soil-HONO impact on the results?

References

Ayres, B. R., Allen, H. M., Draper, D. C., Brown, S. S., Wild, R. J., Jimenez, J. L., Day, D. A., Campuzano-Jost, P., Hu, W., de Gouw, J., Koss, A., Cohen, R. C., Duffey, K. C., Romer, P., Baumann, K., Edgerton, E., Takahama, S., Thornton, J. A., Lee, B. H., Lopez-Hilfiker, F. D., Mohr, C., Wennberg, P. O., Nguyen, T. B., Teng, A., Goldstein, A. H., Olson, K., and Fry, J. L.: Organic nitrate aerosol formation via NO3 + biogenic volatile organic compounds in the southeastern United States, Atmos. Chem. Phys., 15, 13377-13392, https://doi.org/10.5194/acp-15-13377-2015, 2015.

R. Oswald, T. Behrendt, M. Ermel, D. Wu, H. Su, Y. Cheng, C. Breuninger, A. Moravek, E. Mougin, C. Delon, B. Loubet, A. Pommerening-Röser, M. Sörgel, U. Pöschl, T. Hoffmann, M.O. Andreae, F.X. Meixner, I. Trebs: HONO Emissions from Soil Bacteria as a Major Source of Atmospheric Reactive Nitrogen, Science, Vol. 341, Issue 6151, pp. 1233-1235, DOI: 10.1126/science.1242266 , 2013

---

## Author Comment (AC2) · 11 Jan 2018

**Response to Reviewer 2**

We thank the reviewer for their helpful comments.

*Romer et al. disentangles the impact of different processes affecting the O3-T relationship in South Eastern US. The hypothesis and the arguments in the manuscript are well presented and provide robust evidence of the importance of soil-NOx for continental O3 production. Discussion of the results and their implications is scientifically sound. The manuscript should be published in ACP. I only have two minor comments that I would like the authors to address.*

*Minor comments*

*1. At page 9 lines 3-4 the loss of NOx due to NO2 + O3 reaction is taken into account to extract the increase in NOx due to soil emissions. I wonder how much of a change would accounting for the NO2 + NO3 reaction which has a five order of magnitude higher rate constant. I expect no NO3 measurements for the CTR SEARCH network but for the SOAS measurements (Ayres et al. 2015) it should be possible.*

Ayres et al. 2015 found that concentrations of $NO_3$ were extremely low during SOAS and that $N_2O_5$ chemistry was a negligible contributor to $NO_x$ loss (Ayres et al., 2015, Fig. 4). Therefore, the $NO_2+O_3$ reaction rate is equal to the total nighttime $NO_x$ loss. We have revised the section to explain this reasoning:

"To account for the chemical removal of $NO_x$, the cumulative loss of $NO_x$ during the night was added to the observations. During SOAS, the nighttime loss of $NO_x$ occurred almost exclusively through the reaction of $NO_2$ with $O_3$ to form $NO_3$, which then reacted with a VOC to form an organic nitrate (Ayres et al., 2015). $N_2O_5$ chemistry made a negligible contribution to total $NO_x$ loss. The loss rate of $NO_x$ during the night was therefore calculated as the rate of reaction of $NO_2$ with $O_3$. "

*2. The authors are only concerned with soil-NOx emissions although it is now known that soil bacteria are a comparable source of HONO (Oswald et al. 2013). HONO was measured during SOAS (https://data.eol.ucar.edu/dataset/373.037) and its impact on PO3-T is likely convoluted in the 60% contribution of PHOx shown in Fig. 6. In the manuscript it is stated that PHOx is mainly driven by increased solar radiation without showing (or explicitely pointing to) relevant data. However, soil-HONO emissions might also contribute to the PHOx category in Fig. 6. Could the authors attempt a sensitivity analysis or at least discussion of the soil-HONO impact on the results?*

Oswald et al. 2013 found that soil HONO emissions required dry soils, and were enhanced by alkali environments. Neither of these conditions were true during SOAS, and therefore soil HONO emissions are likely negligible at this location. However, when considering ozone-temperature relationships in other locations, the effects of soil HONO emissions should definitely be considered. We have added a discussion of this effect, as well as further explanation of how we concluded that $PHO_x$ was driven by increased solar radiation.

"In very wet environments, soil microbes typically emit $N_2O$ or $N_2$ instead of $NO_x$, and in arid environments soil emissions of HONO can be equal to or larger than soil $NO_x$ emissions (Oswald et al., 2013). Although conditions at the CTR site are too wet and acidic for soil HONO emissions to be significant, in environments where soil HONO emissions are large, they would likely have an even greater effect on ozone production by acting as a source of both $NO_x$ and $HO_x$ radicals."

"The increase in $PHO_x$ with temperature is most likely caused by changes in solar radiation, which is well correlated with the total $PHO_x$ rate (Fig. S7a) and increases strongly with temperature. In contrast, water vapor is not correlated with total $PHO_x$ (Fig. S7b). "

---

## Author Response (AR1)

**Response to Reviewer 1**

**We thank the reviewer for their helpful comments.**

This manuscript is well-written, within the scope of ACP, and provides valuable evidence for the increasing importance of emissions of  $NO_x$  from soils to ozone production as temperatures increase. This manuscript should be published after minor revisions detailed below.

**General Comments**

1. The finding of increased soil  $NO_x$  emission with temperature is valuable, and could be strengthened by a discussion of any known limitations on this effect, such as soil moisture or nitrogen availability. The authors discuss this briefly on page 12, but a more thorough discussion of what is known about microbes would be a valuable addition to this manuscript.

We have expanded the discussion in our paper to discuss additional factors that affect soil  $NO_x$  emissions, including moisture, nitrogen availability, soil type, and pH:

" The only plausible source of  $NO_x$  that matches all of these constraints is soil microbial emissions near to the SOAS site. Soil  $NO_x$  emissions also depend on the water content and nitrogen availability, neither of which is generally limiting in the southeastern United States (e.g., Hickman et al., 2010)."

"Besides temperature, the most important variables affecting soil NOx emissions are typically nitrogen availability and soil water content, as well as plant cover and soil pH (Pilegaard, 2013). In very wet environments, soil microbes typically emit N2O or N2 instead of NOx, and in arid environments soil emissions of HONO can be equal to or larger than soil NOx emissions (Oswald et al., 2013). Although conditions at the CTR site are too wet and acidic for soil HONO emissions to be significant, in environments where soil HONO emissions are large, they would likely have an even greater effect on ozone production by acting as a source of both NOx and HOx radicals.

The variability between sites and the interaction between several biotic and abiotic factors make it difficult to apply regional or model estimates of soil NOx emissions to a particular location. Our approach from this study, using observations of the nighttime atmosphere to determine the NOx emissions rate, helps span the gap between soil chambers and the regional atmosphere. Although soil NOx emissions depend on several environmental factors, process-driven models predict that the

response of soil  $NO_x$  emissions to global warming will be driven primarily by the increase in temperature (Kesik et al., 2006)."

2. The authors should improve the discussion of the effects of local meteorology on surface ozone. Jacob and Winner (2009) also discusses the strong positive relation- ship of ozone with temperature due to the association of temperature with regional stagnation. Was there stagnation on warmer days that would be a contributing factor to the ozone-temperature relationship? Convince the reader of the extent of the effect of the increased  $NO_x$  emissions on this relationship in the context of likely different meteorology on hot days.

Fully disentangling the effects of meteorology, chemistry, and emissions on surface ozone is an open problem, and one that is difficult to answer with in-situ measurements. We have added greater discussion of the possible effects of stagnation on ozone, as well as an indirect calculation of the contribution of non-chemical effects to the ozone temperaturerelationship by comparing how the chemical production, chemical loss, and concentration of ozone all vary during SOAS:

"Regional stagnation episodes, often associated with elevated temperatures, allow ozone to accumulate over several days and are known to contribute significantly to the ozone-temperature relationship (Jacob et al., 1993). How various temperature-dependent chemical effects interact and their relative contributions to ozone production are not well understood outside of polluted environments."

"While elevated temperatures are associated with enhanced production of ozone, they are also associated with increased chemical loss. The chemical loss of ozone occurs through three main pathways in this region: photolysis followed by reaction with H2O, reaction with HO2, and reaction with VOCs (Frost et al., 1998). The loss of O3 was calculated for each of these pathways, and then integrated over the course of the day to determine total daily ozone loss ( $\int LO_3$ ). Chemical loss of ozone is found to increase with temperature (1.1 ± 0.3 ppb ° C-1, Fig. 3b), but much less than the chemical production.

The difference between the trend in the net chemical production and loss of  $O_3$  and the trend in ozone concentration gives a rough estimate of how non-chemical processes contribute to the ozone-temperature relationship. We calculate that nonchemical processes cause  $O_3$  to increase by  $1\pm1.2$  ppb °C-1. This approach does not take into account the interactions between chemical and non-chemical effects, such as how changes to advection and mixing may impact concentrations of VOCs,  $NO_x$ , and other reactants. Although the large uncertainty does not allow for quantitative analysis, qualitatively, chemical and non-chemical processes are both found to be important contributors to the ozone-temperature relationship. Other approaches, such as chemical transport models, that can more directly investigate and control specific physical processes are likely to be better suited to calculating the contribution of non-chemical processes to the ozone-temperature relationship (e.g., Fu et al., 2015). "

3. A final valuable addition would be a statement about whether the authors observe any breakdown of the observed ozone-T relationship at the highest temperatures, as found by Shen et al, 2016 (GRL), or whether their approach could be applied to this problem as well or would be impacted by this phenomenon.

We thank the reviewer for bringing this phenomenon to our attention. Following Shen et al., 2016, we have examined the relationship between ozone and temperature, as well as between ozone production and temperature, in the 5% hottest days from June-August 2010-2014. We find no significant differences between the trend in the top 5% of temperatures and the bottom 95% percent. We have revised the paper to include a mention of this phenomenon, and the results from our analysis:

"A few studies have also reported that this effect is non-linear and can become significantly less strong at the highest temperatures (Steiner et al., 2010; Shen et al., 2016)."

"Based on the long-term SEARCH record, we do not find evidence that the relationship between ozone concentration or ozone production changes significantly at the highest temperatures (the top 5% of observations). This agrees broadly with Shen et al. (2016), who found that ozone suppression at extreme temperatures to be uncommon in the southeastern United States. "

Page 2, line 17-18 – Could you clarify the point of Berlin et al, 2013? They are talking about 'background' ozone coming in to Houston, and I don't see the connection between your point about rural ozone and this paper.

The analysis of Berlin et al., 2013 is indeed about regional background ozone coming in to Houston. When air is entering Houston from the north or northeast, this regional background will contain a component of ozone from rural areas and is associated with greater probability of regulatory exceedances in the Houston area. We have revised this sentence to more accurately reflect the results of Berlin et al., 2013: "From a regulatory perspective, elevated regional background ozone can strongly exacerbate ozone pollution and the probability of regulatory exceedances in urban areas such as Houston (Berlin et al., 2013)."

Page 4, line 23-24 - You say, "When  $HNO_3$  is the most important  $NO_x$  loss pathway,  $O_3$  production and  $NO_x$  loss occur through separate channels and can change independently." Can you clarify this? Aren't both pathways competing for  $NO_2$ , so they are not actually independent? In the example that follows, more explicit statements of what is happening would be useful.

We have revised this section emphasize our point that the relative importance of these two channels can vary, rather than that ozone production and  $NO_x$  loss are strictly independent:

" As the concentration of  $NO_x$  decreases and VOCR increases, the fraction of  $NO_x$  loss that takes place via HNO3 chemistry decreases and the OPE increases (Fig. 2c). The relative importance of HNO3 and RONO2 chemistry determines the relationship between PO3 and LNOx. When HNO3 is the most important NOx loss pathway, O3 production and NOx loss occur through separate channels. O3 production occurs when OH reacts with a VOC, generating RO2 and HO2 radicals; NOx loss primarily occurs when OH reacts with NO2. Although these channels are linked by a shared dependence on OH, the relative importance of these pathways can vary. For example, under these conditions an increase in VOCR will cause NOx loss to decrease, ozone production to increase, and OPE to increase (Fig. 2b– c).

In contrast, when RONO2 chemistry dominates NOx loss, ozone production and NOx loss are intrinsically linked by their shared dependence on the RO2 + NO reaction. This reaction produces O3 in its main channel and consumes NOx in the minor channel that forms organic nitrates, with the ratio between these two channels set by  $\alpha_{eff}$ . Under these conditions, changes to the chemistry that do not affect  $\alpha_{eff}$  have a minimal effect on OPE (Fig. 2d) and the OPE can be considered to be unvarying with temperature. An increase in VOCR or a decrease in NOx will affect both NOx loss and ozone production equally, because both processes are dependent on the same set of reactions. Because of this change in behavior, from variable OPE to fixed OPE, the drivers of the O3-T relationship are expected to be categorically different in areas where RONO2 chemistry dominates NOx loss, are unlikely to apply in areas with low concentrations of NOx and high concentrations of reactive VOCs, where RONO2 chemistry is most important. In these areas, more NOx must

be oxidized in order to produce more O3. "

Page 4, line 30 - It is unclear to me whether you include thermal decomposition of PAN for example here, so that if temperature goes up, the effective yield of the sink would go down and OPE would not be fixed.

We do not include PAN in these calculations. PAN is quite short lived under typical conditions of the CTR site, and therefore does not serve as a permanent sink of  $NO_x$ . Furthermore, it has been found at multiple forested locations that total peroxy nitrate concentrations do not vary significantly with temperature, due to changes in both production and loss. As a result, the effect of PAN on OPE is likely to be small. We have added a paragraph to section 2 explaining this reasoning:

" NOx also has several temporary sinks that can sequester NOx, most importantly peroxy acyl nitrate (PAN). In the summer-time southeastern United States, the lifetime of PAN is typically 1–2 hours, too short to act as a permanent sink of NOx. Past studies in forested regions have found remarkably little variation in PAN with temperature, due to compensating changes in both its production and loss (e.g., LaFranchi et al., 2009). As a result, the formation or destruction of PAN does not contribute significantly to net ozone production or NOx loss and we do not include it in these calculations. "

**Also, if you are integrating over a day, do you think that ignoring deposition is at all important?**

Since deposition of  $NO_x$  is far slower than its chemical removal, deposition will only affect OPE if it affects the fraction of  $NO_x$  sinks that recycle or remove  $NO_x$  from the atmosphere. Deposition is not a major sink for any of the species that can recycle  $NO_x$  to the atmosphere and therefore changes in deposition with temperature are unlikely to be important. We have expanded our discussion of  $RONO_2$  chemistry to explain this effect:

"Deposition is only a minor loss process for  $\Sigma RONO_2$ , therefore any changes in the deposition rate with temperature will have at most a minor effect on  $\eta$ ."

**Page 7, line 18 - How does an average OPE of 45 compare to OPE calculated from your model of $PO_3/LNO_x$ ?**

To better constrain our understanding of OPE, we compare the OPE calculated from the ratio of  $\int PO_3$  and  $\int LNO_x$  to the OPE calculated as the ratio of  $O_x$  to  $NO_z$ . These two different calculations of OPE agree reasonably well, bolstering our confidence in the calculated ozone production and  $NO_x$  loss rates:

" OPE can also be estimated from the ratio of odd oxygen ( $O_x \equiv O_3 + NO_2$ ) to  $NO_x$  oxidation products ( $NO_z \equiv NO_y - NO_x$ ) (Trainer et al., 1993). The afternoon ratio of  $O_x$  to  $NO_z$  during SOAS varied from 43–67 (interquartile range), slightly higher than the average ratio of  $\int PO_3$  to  $\int LNO_x$ . However, since the  $O_x$  to  $NO_z$  ratio includes the effects of chemical loss and transport, which the ratio of  $\int PO_3$  to  $\int LNO_x$  does not, these two values are not expected to be equivalent, particularly in non-polluted areas. "

Page 7, line 19 – Why do you say there is no OPE trend, but then provide a value (0.2)? If it is not statistically significant, don't show a number.

We think that the calculated trend and error in OPE with temperature provide useful information even though the trend is not statistically significant. Because both the calculated trend and error are close to zero, we can be confident that OPE does not change dramatically with temperature, even though the trend is not significantly different from zero. We have revised this sentence to emphasize the result that OPE is found not to vary with temperature rather than the statistical significance of the result:

"As expected from the importance of RONO2 chemistry to NOx loss,  $\int LNO_x$  and  $\int PO_3$  are tightly correlated (r2 = 0.90), and OPE is high (OPE average 45±3 ppb ppb-1) and is effectively constant with temperature (calculated trend 0.2±0.6 °C-1). Therefore, the increase in  $\int PO_3$  with temperature is not caused by more efficient production of ozone while the same amount of NOx is consumed."

Page 10, line 20 - You say, "The increase of  $PHO_x$  is mostly driven by increased solar radiation, and not by temperature directly." Could it not also be driven by increased water vapor with higher temperatures?

While water vapor is a major reactant in  $HO_x$  radical production, we find that its contribution to the increase of  $PHO_x$  with temperature to be minimal at this location. Water vapor is effectively constant with

temperature and is not correlated with the total  $PHO_x$  rate, while solar radiation increases with temperature and is well correlated with the total  $PHO_x$  rate. Because of this difference, we are confident in assigning the change in  $PHO_x$  to variation in solar radiation. We have expanded the discussion of  $PHO_x$  and added a figure to the supporting information to clarify our reasoning:

"The increase in  $PHO_x$  with temperature is most likely caused by changes in solar radiation, which is well correlated with the total  $PHO_x$  rate (Fig. S7a) and increases strongly with temperature. In contrast, water vapor is not correlated with total  $PHO_x$  (Fig. S7b). "

Page 12, line 22 – You say "These emissions cannot be regulated or controlled directly, and therefore present challenges to traditional air quality management techniques." Then this statement seems to be a contradiction - "Alternative approaches, such as changes to fertilizer application practices, have the potential to significantly reduce SNOx from agricultural regions (Oikawa et al., 2015)."

We have removed the statement that soil  $NO_x$  emissions cannot be regulated or controlled directly, and instead emphasize that there are additional difficulties associated with controlling sources of  $NO_x$ distributed over large areas:

"Because these emissions are distributed over broad areas and are not directly anthropogenic, they present additional challenges to air quality management. Indirect approaches, such as changes to fertilizer application practices, have the potential to significantly reduce  $S_{NO_x}$  from agricultural regions (Oikawa et al., 2015). "

**Response to Reviewer 2**

**We thank the reviewer for their helpful comments.**

Romer et al. disentangles the impact of different processes affecting the O3-T relationship in South Eastern US. The hypothesis and the arguments in the manuscript are well presented and provide robust evidence of the importance of soil-NOx for continental O3 production. Discussion of the results and their implications is scientifically sound. The manuscript should be published in ACP. I only have two minor comments that I would like the authors to address.

**Minor comments**

1. At page 9 lines 3-4 the loss of NOx due to NO2 + O3 reaction is taken into account to extract the increase in NOx due to soil emissions. I wonder how much of a change would accounting for the NO2 + NO3 reaction which has a five order of magnitude higher rate constant. I expect no NO3 measurements for the CTR SEARCH network but for the SOAS measurements (Ayres et al. 2015) it should be possible.

Ayres et al. 2015 found that concentrations of NO3 were extremely low during SOAS and that N2O5 chemistry was a negligible contributor to NOx loss (Ayres et al., 2015, Fig. 4). Therefore, the NO2+O3 reaction rate is equal to the total nighttime NOx loss. We have revised the section to explain this reasoning:

"To account for the chemical removal of NOx, the cumulative loss of NOx during the night was added to the observations. During SOAS, the nighttime loss of NOx occurred almost exclusively through the reaction of NO2 with O3 to form NO3, which then reacted with a VOC to form an organic nitrate (Ayres et al., 2015). N2O5 chemistry made a negligible contribution to total NOx loss. The loss rate of NOx during the night was therefore calculated as the rate of reaction of NO2 with O3. "

2. The authors are only concerned with soil-NOx emissions although it is now known that soil bacteria are a comparable source of HONO (Oswald et al. 2013). HONO was measured during SOAS (https://data.eol.ucar.edu/dataset/373.037) and its impact on PO3-T is likely convoluted in the 60% contribution of PHOx shown in Fig. 6. In the manuscript it is stated that PHOx is mainly driven by increased solar radiation without showing (or explicitly pointing to) relevant data. However, soil-HONO emissions might also contribute to the PHOx category in Fig. 6. Could the authors attempt a sensitivity analysis or at least discussion of the soil-HONO impact on the results?

Oswald et al. 2013 found that soil HONO emissions required dry soils, and were enhanced by alkali environments. Neither of these conditions were true during SOAS, and therefore soil HONO emissions are likely negligible at this location. However, when considering ozone-temperature relationships in other locations, the effects of soil HONO emissions should definitely be considered. We have added a discussion of this effect, as well as further explanation of how we concluded that PHOx was driven by increased solar radiation.

"In very wet environments, soil microbes typically emit  $N_2O$  or  $N_2$  instead of  $NO_x$ , and in arid environments soil emissions of HONO can be equal to or larger than soil  $NO_x$  emissions (Oswald et al., 2013). Although conditions at the CTR site are too wet and acidic for soil HONO emissions to be significant, in environments where soil HONO emissions are large, they would likely have an even greater effect on ozone production by acting as a source of both  $NO_x$  and  $HO_x$  radicals."

"The increase in  $PHO_x$  with temperature is most likely caused by changes in solar radiation, which is well correlated with the total  $PHO_x$  rate (Fig. S7a) and increases strongly with temperature. In contrast, water vapor is not correlated with total  $PHO_x$  (Fig. S7b). "

**Effects of temperature-dependent $\mathbf{NO}_{\mathbf{x}}$ emissions on continental ozone production**

Paul S. Romer1, Kaitlin C. Duffey1, Paul J. Wooldridge1, Eric Edgerton2, Karsten Baumann2,3, Philip A. Feiner4, David O. Miller4, William H. Brune4, Abigail R. Koss5,6,7, Joost A. de Gouw6,7, Pawel K. Misztal8, Allen H. Goldstein8,9, and Ronald C. Cohen1,10

[revised manuscript text omitted]

NOx cycles (Fig. 2a). HO2 and RO2 radicals are generated in the HOx cycle when an organic compound is oxidized by a VOC reacts with OH in the presence of NOx. In one turn of the cycle, the VOC is oxidized, OH is regenerated, and two molecules of O3 are formed. The reactions that drive these catalytic cycles forward are in constant competition with reactions that remove radicals from the atmosphere, terminating the cycles. Termination can occur either through the association of two HOx radicals to form nitric acid or an organic nitrate.

The balance between propagating and terminating reactions causes  $PO_3$  to be a non-linear function of the NOx and VOC reactivity (VOCR), as well as the net-production rate of HOx radicals ( $PHO_x$ ). Net sources. The largest source of HOx radicals

include photolysis of species such as  $O_3$  and formaldehyde in the summertime is the photolysis of  $O_3$  followed by reaction with water vapor to produce OH; additional sources include the photolysis of formaldehyde and peroxides, ozonolysis of alkenes, and isomerization pathways in the oxidation of isoprene and other VOCs. To understand the response of ozone production to changes in chemistry, we use a simplified framework based on the balance of  $HO_x$  radical production and loss (Farmer et al., 2011).

5

Under high or moderate  $NO_x$  conditions, the primary loss process of  $HO_2$  and  $RO_2$  radicals is reaction with NO and the concentration of OH radicals can be expressed as a quadratic equation. To modify this approach to work under low- $NO_x$  conditions, reactions between  $HO_x$  radicals must also be included, leading to a set of 4 algebraic equations that can be solved numerically (details given in Appendix A). Fig. Figure 2b shows the calculated rate of ozone production as a function of  $NO_x$

10 at two different VOC reactivities. Depending on atmospheric conditions, the ozone production rate can either be  $NO_x$ -limited, where additional  $NO_x$  causes  $PO_3$  to increase, or  $NO_x$ -saturated, where additional  $NO_x$  suppresses ozone formation.

When considering day-to-day variations, the total amount of ozone produced over the course of a day ( $\int PO_3$ ) is a more representative metric than the instantaneous ozone production rate. Total daily ozone production depends on all of the factors that affect  $PO_3$  as well as their diurnal evolution. In places where ozone production is NOx-limited, changes to chemistry with tem-

[revised manuscript text omitted]